# Energy (and Reactive Oxygen Species Generation) Saving Distribution of Mitochondria for the Activation of ATP Production in Skeletal Muscle

**DOI:** 10.3390/antiox12081624

**Published:** 2023-08-17

**Authors:** Alejandra Espinosa, Mariana Casas, Enrique Jaimovich

**Affiliations:** 1Center for Studies of Exercise, Metabolism and Cancer (CEMC), Instituto de Ciencias Biomédicas, Facultad de Medicina, Universidad de Chile, Santiago 8320000, Chile; alejandra.espinosa@uv.cl (A.E.);; 2San Felipe Campus, School of Medicine, Faculty of Medicine, Universidad de Valparaiso, San Felipe 2172972, Chile

**Keywords:** ATP production, mitochondrial network, mitochondria dynamics, MCU

## Abstract

Exercise produces oxidants from a variety of intracellular sources, including NADPH oxidases (NOX) and mitochondria. Exercise-derived reactive oxygen species (ROS) are beneficial, and the amount and location of these ROS is important to avoid muscle damage associated with oxidative stress. We discuss here some of the evidence that involves ROS production associated with skeletal muscle contraction and the potential oxidative stress associated with muscle contraction. We also discuss the potential role of H_2_O_2_ produced after NOX activation in the regulation of glucose transport in skeletal muscle. Finally, we propose a model based on evidence for the role of different populations of mitochondria in skeletal muscle in the regulation of ATP production upon exercise. The subsarcolemmal population of mitochondria has the enzymatic and metabolic components to establish a high mitochondrial membrane potential when fissioned at rest but lacks the capacity to produce ATP. Calcium entry into the mitochondria will further increase the metabolic input. Upon exercise, subsarcolemmal mitochondria will fuse to intermyofibrillar mitochondria and will transfer the mitochondria membrane potential to them. These mitochondria are rich in ATP synthase and will subsequentially produce the ATP needed for muscle contraction in long-term exercise. These events will optimize energy use and minimize mitochondria ROS production.

## 1. Introduction

Since the first report that muscular physiology is redox-dependent, several research lines in redox biology have grown significantly. Skeletal muscle is the human buffer against aging, some metabolism pathologies, and neuropathologies, so the understanding of how skeletal physiology processes are modulated by reactive oxygen species (ROS) is a source of potential intervention based on diet supplementation or exercise protocols. Evidence suggests that exercise produces oxidants from a variety of intracellular sources, including NADPH oxidases (NOX) and mitochondria as the main sources [1,2]. Exercise-derived ROS are beneficial, and the amount and location of these ROS is important to avoid muscle damage associated with oxidative stress. In this review, we propose a model in which the role of the fusion of different subpopulations of mitochondria after exercise is relevant, increasing energy saving and decreasing H_2_O_2_ production.

## 2. An Energy-Saving Mechanism in Skeletal Muscle

### 2.1. Mitochondria Distribution and Characteristics in Skeletal Muscle

Skeletal muscle fibers can change from a preferentially glycolytic metabolism to oxidative metabolism upon certain types of physiological exercise, a process for which the cellular mechanism needs to be better understood. In turn, the excitation–contraction (EC) mechanism is connected to a network of mitochondria in skeletal muscle fibers, showing a physiological communication between contractile machinery and mitochondria, in which Ca^2+^ has a relevant role. Moreover, mitochondria are mobile and plastic organelles, constantly changing shape, fusing, or fissioning with each other, and changing their role in parallel in cellular bioenergetics [3].

Adult skeletal muscle presents two populations of mitochondria: one group comprises both perinuclear (PN) and perivascular (PV) mitochondria, both considered peripherally located mitochondria (PLM) or subsarcolemmal mitochondria (SSM); the other group is called intermyofibrillar (IMF) mitochondria (see Figure 1) [4]. The SSM population comprises the mitochondria located beneath the plasma membrane of the muscle fiber (and, as nuclei have a similar location in adult skeletal muscle, mitochondria closely surrounding the myonuclei) and the IMF population of mitochondria, located regularly close to the sarcoplasmic reticulum terminal cisternae and the triad (the grouping of two terminal cisternae an one transverse tubule), at regular intervals along every myofibril [5,6].

IMF mitochondria form a structural arrangement characterized by the interaction of transverse mitochondrial tubules in the sarcomere, called “mitochondrial reticulum”, which has been proposed as an energetic conductive pathway from mitochondria to the contractile apparatus [4]. Specialized proteins play roles such as intermembrane linkers, which are formed by a single protein with two membrane-interacting domains in the sarcoplasmic reticulum [8]. This physical proximity, such as voltage-dependent anion channel (VDAC) and RyR1 or IP3R, allows the passage of calcium from the reticulum to the mitochondria to be more effective. It has also been described that H_2_O_2_ would diffuse from the mitochondrial space towards these contact domains to modulate Ca^2+^ release locally [8,9]. It is very interesting to note that these two populations of mitochondria with different membrane potential not only contribute both to muscle fiber oxidative capacity and bioenergetics but also do so each in a specialized way, allowing in the end, to optimize ATP production precisely where is needed for contraction, i.e., near the myofibrils, by producing mitochondria membrane potential propagation. In agreement with this, it has been proposed that SSM supports the IMF energy, based on the presence of higher oxidative enzyme activity [10].

Together with the different localization and oxidative capacity, both populations express different types of proteins and different membrane potentials. For example, SSM expresses mitochondrial calcium uniporter regulator 1 (MICU1) [11]. Mitochondrial calcium uniporter (MCU) is a highly selective and highly regulated calcium channel inserted in the inner mitochondrial membrane that allows calcium uptake from the cytosol to mitochondria after contraction [10,12]. This complex (when regulated by MICU1) is basally closed and is activated upon cytoplasmic Ca^2+^ increases, allowing then the ion to enter the mitochondria matrix. ROS production is normally high when the mitochondrial potential (ΔΨ) is elevated. Therefore, MICU1 expression, by regulating Ca^2+^ entry, would protect against mitochondrial H_2_O_2_-generated damage [13].

This idea is consistent with the evidence showing that, in neurons, MCU promoted the activity of the electron transport chain and the chemical reduction of NAD+ to NADH, which would imply that electrons are not available for ROS formation [14]. Moreover, it is important to highlight that high MCU activity also induces mitochondrial membrane potential dissipation, facilitating the activity of the ATP synthase [15]. It has been observed that the lower expression of MCU generates muscle atrophy, intimately relating the regulation of mitochondrial Ca^2+^ to the size of the muscle fiber [16]. The action of MCU is crucial for ATP synthesis in aerobic conditions because Ca^2+^ increase is an essential cofactor for the tricarboxylic acid (TCA) cycle’s enzymes such as glycerol phosphate dehydrogenase (GPDH), pyruvate dehydrogenase (PDH), isocitrate dehydrogenase (ICDH), and α-ketoglutarate dehydrogenase (α-KGDH) [17]. Thus, Ca^2+^ increase can elevate the efficiency of complexes I, III, and IV of OXPHOS. The Ca^2+^-induced activation of mitochondrial sodium–calcium exchanger (NCLX) results in a Na^+^ influx into the matrix, and Na^+^ interacts with phospholipids in the inner leaflet of the inner membrane, decreases its fluidity, and slows down ubiquinol (UQH2) diffusion, increasing electron flux due to major NADH availability. In simple words, calcium potentiates the establishment of the proton gradient needed for ATP synthesis, which is used for sarco/endoplasmic reticulum Ca^2+^-ATPase (SERCA) to pump back calcium from the cytosol to the SR [18].

When we talk about muscle activity, it is important to remember that there are different types of muscle fibers (as different types of motor units). Slow, fatigue-resistant fibers possess a larger number of mitochondria, having a preferential oxidative metabolism that allows them to contract for long periods of time. Fast, fatigable fibers have lower mitochondria content, relying on an anaerobic glycolytic metabolism, which is responsible for their poor fatigue resistance. These different types of muscle fibers can change from one phenotype to another depending on external demands. One of the main actors in maintaining or changing the muscle phenotype is the pattern of stimulation coming from motor neurons. In this way, it has been shown that low-frequency electrical stimulation induces the expression of genes belonging to the slow phenotype, while high frequencies lead to the expression of genes typical for fast phenotypes. Among the slow phenotype transcriptional profile, we can find slow isoforms of proteins from the contractile apparatus and of factors that lead to increased mitochondrial biogenesis as well as oxidative enzymes [19,20]. Interestingly, Quezada et al. have also shown that low-frequency electrical stimulation (which allows fibers to convert to slow-phenotype fibers) can decrease the expression of the MCU complex in isolated adult fibers [21].

Higher protein levels of MCU and MICU1 per mitochondrion have been observed in fast-phenotypes muscles (like *flexor digitorium longus*) than in slow phenotype muscles, such as the soleus. From these data, we can propose a hypothesis in which the decrease in mRNA of the MCU complex after the low-frequency electrical stimulation of isolated fibers from a fast muscle would favor a lower protein level of MCU and MICU1 per mitochondrion, being an early metabolic response to the phenotypic shift from fast to slow phenotype muscle fiber [22]. Also, a gradual increase in the number of mitochondria, together with a decrease in levels of the MCU complex in response to a low-frequency electrical stimulus, could allow adapting mitochondrial Ca^2+^ homeostasis to finally reach that of a slow muscle. On the other hand, *Mcu* gen deletion produces a decrease in Ca^2+^-stimulated ATP synthesis, impairment in TCA cycle substrate flux, and a turn toward fatty acid metabolism [12].

These data suggest that both mitochondria calcium transients and the total volume of mitochondria are somehow conjointly modulating metabolism to provide either a fast-fatigable or a slow-fatigue resistant response.

### 2.2. Function of the Heterogeneity of Mitochondria within the Muscle Fiber

The SS mitochondria are quite different to the IMF mitochondria [7,11,23]. It has been shown that electron transport chain elements needed to establish the mitochondria membrane potential are differentially distributed among SSM and IMF; in particular, complex IV and cytochrome C are located mostly in the SSM [11,23]. There are also differences in the distribution of complex V (ATP synthase), which is located mostly in the IMF mitochondria [23]. In contrast, the importin translocase of the outer membrane, TOM20, has homogeneous distribution in all the cell’s mitochondria, suggesting a specific compartmentalization of different mitochondrial proteins between mitochondria subpopulations [11].

This evidence, together with the fact that there is a shift in mitochondria membrane potential (higher in SSM in resting conditions) towards the center of the muscle fiber upon electrical stimulation [11], prompted us to propose that, in the SSM pool, the main proton motive force is generated by the activation of complexes I-IV. In contrast, ATP is generated in complex V, which is in the IMF mitochondria, and this ATP production will occur only after the fusion of both mitochondria populations with the consequent spreading of mitochondria membrane potential. As mitochondria membrane potential is established across the inner mitochondria membrane, it can only propagate via the fusion of mitochondria membranes. Mitochondrial ROS production will be minimal since the proton motive force is used for ATP synthesis when the electron transport chain works at a maximum level induced by exercise.

In fact, *Mcu* deletion produces a decrease in Ca^2+^-stimulated ATP synthesis, impairment in TCA cycle substrate flux, and a turn toward fatty acid metabolism [12]. Stimuli, such as extracellular ATP or electrical stimulation, can increase the expression of MCU in isolated adult fibers [21]. In turn, it has been shown that the MCU-dependent increase in mitochondrial ROS is necessary for optimal skeletal muscle repair after an injury via the induction of actin polymerization dependent on RhoA [24]. It is interesting to note that, using the direct measurement of superoxide via electron paramagnetic resonance, Crochemore et al. demonstrated that SSM produce more superoxide than IMF mitochondria [25].

In summary, skeletal muscle mitochondria achieve this performance by separating two important functions: the first is mitochondria membrane potential generation via calcium-sensitive oxidative phosphorylation (which occurs mainly in the subsarcolemmal population of mitochondria), and the second function is ATP production mediated by ATP synthase, which occurs in the intermyofibrillar population of mitochondria. To achieve this amazing performance, the two populations of mitochondria are not connected at rest and have a different resting protein composition. The intermyofibrillar mitochondria are enriched in ATP synthase and have a high content of mitochondrial calcium uptake 1 (MICU1), MICU1, as a key regulator of mitochondrial Ca^2+^ uptake, which negatively regulates calcium entry into the mitochondrial matrix through the MCU calcium channel [26,27]. This protein is a Ca^2+^ sensor, and its functioning does not impact ΔΨ or oxygen consumption [26,28]. On the other hand, the subsarcolemmal mitochondria are enriched in the electron transport chain complex proteins and, having no MICU1, can reach a high calcium content upon muscle activation (Figure 2). The model in Figure 2 shows that exercise induces the mitochondrial fusion of SSM and IMF, transferring electrical properties and proteins from SSM to the mitochondrial network, decreasing the ROS generation, and improving energy-saving design. This process is reversible, and, consequently, in resting conditions after exercise, the fission of mitochondria will occur, and the high mitochondria membrane potential in SSM can be recovered.

### 2.3. Mitochondria Dynamics Are Altered in Skeletal Muscle of Aging Subjects and in Pathological Conditions

Mitochondrial fusion events in skeletal muscle fibers are hard to evidence due to the highly restricted space in which IMF mitochondria are located; fusion events nevertheless take place, and they were shown for the first time by Eisner et al. in 2014 [29].

When we consider the above-described model, it is reasonable to assume that mitochondria dynamics (fission and fusion) play an essential role in skeletal muscle function and wellbeing. Maintaining optimal skeletal muscle health requires dynamic mitochondrial function, which becomes disrupted in aging and various pathological conditions. Age-related mitochondrial dysfunction is characterized by impaired fusion/fission processes and mitophagy, leading to sarcopenia and reduced exercise capacity. In addition, diseases such as muscular dystrophies, mitochondrial myopathies, and type 2 diabetes exhibit mitochondrial fusion and fission imbalances, contributing to impaired excitation–transcription coupling (ETC, see Figure 3).

Alterations in mitochondria dynamics have been shown to occur in middle age and more advanced age in mice [30], and they parallel dramatic decreases in muscle function. This was evidenced by changes in mitochondria orientation as seen by confocal microscopy, changes in both mitochondria size and shape as seen by electron microscopy, and changes in the expression of proteins involved in fission and fusion processes. Furthermore, mitochondria dynamics appeared to be altered in the skeletal muscle of a mouse model of alcohol consumption [29]. Oxidative stress and reduced antioxidants reinforce mitochondrial fragmentation, suppress fusion/fission, and impair the electron transport chain, decrease ATP production, and cause DNA damage. Mitochondrial dynamics depend on proteins such as mitofusin (MFN) 1 and 2 and optic atrophy protein 1(OPA1) for the fusion of the outer and inner mitochondrial membranes. Mitochondrial fission involves the recruitment of cytoplasmic Drp1 to the mitochondrial outer membrane, forming a ring-like structure with adaptors Mff, MiD49, and MiD51. As schematized in Figure 3, these protein interactions ensure a balance between fusion and fission, which is crucial for maintaining mitochondrial dynamics and cellular functions [31]. The deletion of OPA1 leads to mitochondrial dysfunction, reduced myogenic stem cells, decreased protein synthesis, and the activation of protein breakdown. *Opa1* deletion in skeletal muscle affects the entire body, causing a premature aging phenotype that ultimately leads to animal death [32]. *Opa1* deficiency in myopathy engages TLR9, activating NF-κB and triggering muscle inflammation. This localized inflammation can become systemic, impacting the entire body. An altered growth hormone/IGF1 axis and enhanced FGF21 expression are observed, contributing to impaired growth. *Opa1* deficiency disrupts growth-related processes and promotes inflammation via inflammatory pathway activation. These processes collectively contribute to the development and progression of myopathy [33]. Aging is associated with sarcopenia, and when obesity is associated, the term sarcopenic obesity (SO) is applied. Dysfunctional adipose tissue, fatty acid excess inside the bloodstream, and low-grade systemic inflammation are combined, resulting in lipotoxicity, oxidative stress, insulin resistance, and inflammation in the skeletal muscle [34]. It has been reported that a high-fat diet suppresses mitochondrial biogenesis in the skeletal muscle of zebrafish and decreases the number of SSM, Mfn2, and OPA1. On the contrary, Fis1 and Drp1, both fission proteins, were found to be increased in an obese mouse model, as opposed to in non-obese conditions in mice [31]. On the other hand, mitochondrial uncoupling attenuates SO by enhancing skeletal muscle mitophagy, reducing muscle inflammation, promoting mitochondrial turnover via STAT3 signaling, and mitigating muscle degradation [35].

Exercise enhances the healthy mitochondrial network, promoting fusion/fission markers and biogenesis. Sarcopenia diminishes mitochondrial dynamics, mitophagy markers, and network efficiency, while exercise stimulates mitochondrial biogenesis via PGC1-α activation. Also, PGC-1α overexpression mitigates age-related increases in mitophagy markers, including Fis1 and Drp1 proteins, improves mitochondrial function, and reduces oxidative damage in mouse muscle [36]. Moderate-intensity exercise can be a non-invasive treatment, activating pathways that regulate the mitochondrial network in skeletal muscle [37]. Reduced MICU3 expression during aging leads to decreased mitochondrial Ca^2+^ uptake, and studies on aged mice and senescent C2C12 cells revealed that MICU3 downregulation is associated with decreased myogenesis, increased ROS, and apoptosis. Restoring MICU3 levels increases antioxidant defenses and promotes myogenesis. These findings highlight that MICU3 is a contributing factor to ROS production and apoptotic processes during aging [38]. In the same line, mitochondrial dynamics, and ROS production play crucial roles in the pathogenesis of Duchenne muscular dystrophy (DMD), a genetic skeletal muscle disorder characterized by mutations in the gene that encodes dystrophin. Studies have shown that DMD patients present disruptions in mitochondrial fusion and fission processes, leading to mitochondrial dysfunction and generating fragmented and dysfunctional mitochondria in muscle fibers. Intracellular Ca^2+^ disruption also has been reported to be derived from both increased Ca^2+^ influx and altered calcium release, leading to abnormally elevated resting cytosolic Ca^2+^ concentration [39]. Consequently, these effects contribute to the apoptosis of muscle cells. These abnormalities increase ROS production and oxidative stress, compromise energy production, and impair cellular signaling, exacerbating muscle weakness and degeneration in DMD. Recent studies have highlighted the involvement of ferroptosis, a form of regulated cell death involving iron-dependent lipid peroxidation, in DMD pathology [40,41]. The dysregulation of NRF_2_ in DMD may promote ferroptosis in muscle cells by contributing to increased susceptibility to cell death. In turn, this can exacerbate muscle degeneration and inflammation observed in DMD.

Some studies have reported cases of statin-induced, muscle-related side effects, including myopathy or rhabdomyolysis. Atorvastatin dose-dependently inhibits C2C12 cell viability, resulting in increased intracellular iron ions, ROS, and lipid peroxidation. These effects primarily occur in mitochondria, leading to mitochondrial dysfunction. Biomarkers of myocardial injury are elevated during atorvastatin treatment, but ferroptosis inhibitors can counteract these effects [42]. Mechanistically, GSH depletion, along with the decrease in Nrf2, GPx4, and xCT cystine-glutamate antiporter, contributes to atorvastatin-induced muscular cell ferroptosis and damage [42]. Moreover, an increase in iron overload, senescence, and muscle atrophy markers was found in old senescence-accelerated mouse-prone 8 (SAMP8), a sarcopenia-like phenotype, suggesting that iron-overload-induced ferroptosis plays an essential role in sarcopenia [43].

## 3. H_2_O_2_-Mediated Physiological Signals in Skeletal Muscle

### 3.1. Role of H_2_O_2_ in the Excitation–Contraction (EC) Coupling Mechanism

In the EC coupling mechanism, there is direct evidence pointing to hydrogen peroxide as the oxidizing agent that modulates the interaction between the Cav.1.1 and the ryanodine-1 receptor (RyR1) and, on the other hand, the oxidation of the contractile apparatus. In EC coupling, Cav1.1 is an L-type Ca^2+^ channel located in the T tubule of skeletal muscle cells, composed of 5 subunits (α1S, α2, β, γ and δ), of which the α1S subunit is the one carrying the transmembrane barrels responsible for its voltage sensor capacity during a change in membrane potential, as occurs during an action potential [44,45]. Cav 1.1 plays a fundamental role in changing its spatial conformation after membrane depolarization, which induces RyR1 channel opening, and the following massive release of Ca^2+^ from the sarcoplasmic reticulum to the cytoplasm, which triggers the contraction of the muscle fiber [46,47,48]. A modulatory role of H_2_O_2_ in skeletal muscle contraction has been suggested for decades, based on the idea that skeletal muscle has many redox-sensitive proteins involved in the contractile apparatus and mitochondria [49]. However, the intracellular H_2_O_2_ concentration in skeletal muscle fibers is relatively low, estimated to be less than 10^−7^ M (100 nM). Contractions induced a smaller increase in intracellular H_2_O_2_ compared to exposure to 10^–6^ M (1 μM) H_2_O_2_ [50]. H_2_O_2_ induces a delay in action potential associated with excitation processes in mouse myoblasts, which was reverted using the antioxidant N-acetylcysteine (NAC) [51]. Indeed, a decrease in the action potential amplitude was observed in H_2_O_2_-treated fibers after high-frequency stimulation with respect to non-treated fibers [52]. The addition of H_2_O_2_ is known to affect EC coupling, but contradictory information is found in the literature. Some reports show that µM doses apparently do not affect EC coupling, but mM doses have an inhibitory effect [53]. On the other hand, 10 mM H_2_O_2_ could increase the Ca^2+^ sensitivity of contractile machinery. Moreover, the effect of 1mM H_2_O_2_ on contractile response also depends on the fiber type: depolarization-induced contraction of slow but not fast twitch fibers [54]. One of the central stimuli that increase H_2_O_2_ production is the electrical stimulation of isolated fibers or cultured muscle cells, an in vitro strategy mimicking physical activity. For example, contraction by the electrical stimulation of myotubes or nude fibers can generate intracellular increase in both H_2_O_2_ and NO [55,56,57].

A particularly interesting protein is SERCA. This calcium pump is responsible for decreasing Ca^2+^ concentrations to allow muscle relaxation. It has been reported that SERCA expression decreases upon oxidative conditions [58].

Exercise could be considered an acute stressor, which at low doses induces H_2_O_2_ production as a “hormetic” signal, inducing a gene expression program of defense against oxidative stress [59,60]. In this line, acute exercise significantly increased mitochondrial H_2_O_2_ and FOXO3a, a key transcription factor involved in the activation of transcription of mitochondrial superoxide dismutase and catalase, both key antioxidants enzymes [28]. The same exercise pattern upregulates redox effector factor-1 (Ref1) and nuclear factor erythroid 2-related factor 2 (Nrf2) and increases reduced glutathione (GSH) content and manganese superoxide dismutase activity, suggesting that a low level of ROS production is a stimulus for antioxidant gene expression. Along the same lines, it has been observed that cultured myotubes incubated with low doses of H_2_O_2_ increase the expression of hemoxygenase 1, an enzyme related to antioxidant protection [34].

The controlled production of ROS exerted by exercise is a factor that allows the muscles to adapt to training, triggering the activation of transcription factors and the expression of genes such as peroxisome proliferator-activated receptor gamma co-activator 1 α (PGC1 α) and peroxisome proliferator-activated receptor γ (PPAR γ), the first of which is related to the induction of mitochondrial myogenesis, while the second is related to the increase in uncoupling protein-3 (UCP3), a mitochondrial uncoupling protein, which is considered the first line of defense against mitochondrial ROS production [13,56]. In addition, the upregulation of UCP3 may help reduce ROS production by increasing the proton gradient and protecting muscle mitochondria from oxidative stress during exercise. However, it also compromises energy coupling efficiency, as indicated by an increased respiration rate [35].

Another mechanism involved in ROS production with skeletal muscle physiology is extracellular ATP release. This mechanism has been described in macrophages where ROS increase triggers the activation of pathways needed for cytokine release [61]. When skeletal muscle fibers are electrically stimulated at low frequencies (20 Hz), ATP is released from the muscle fibers and binds to their P2Y purinergic receptors, thereby activating a signal transduction pathway, which culminates in the regulation of the transcription of some genes by Ca^2+^ release via inositol 1,4,5-trisphosphate receptors (IP_3_R) from the sarcoplasmic reticulum. Indeed, Cav1.1 is also responsible for triggering an excitation–transcription coupling (ETC) process, which is mediated by the release of ATP [62,63,64]. H_2_O_2_ may also have a role in the ETC process, considering that, after contraction, ATP binds to P2Y1 receptors and then activates NOX2 via a protein kinase C (PKC)-dependent mechanism [62]. This evidence shows that exercise results in skeletal muscle’s exposure to H_2_O_2_, which is involved in adaptive physiological responses.

### 3.2. Role of H_2_O_2_ in Glucose Uptake in Skeletal Muscle

Among the physiological functions of H_2_O_2_ in skeletal muscle, in addition to its role as a modulator of EC and ET coupling, metabolic participation also has been described. Muscle contraction is a process with high energy demand, so the EC coupling must have machinery associated with the constant generation of ATP. In addition, it has been shown that the production of ROS by various sources activates the entry of glucose into the cell. Myotubes treated with NAC for 24 h decreased the mRNA and protein contents of glucose transporter type 4 (GLUT4), the mRNA content and activity of phosphofructokinase (PFK), and lactate production and glucose uptake [65]. As for muscle contraction, exogenous stimulation with H_2_O_2_ increased 2-Deoxy-D-glucose uptake, adenosine monophosphate-activated protein kinase (AMPK) [66], and glycolytic activity [67]. These findings suggest that ROS produced after muscle contraction play an important role in increasing glycolytic activity and glucose uptake after exercise. On the other hand, H_2_O_2_ stimulation in high doses is capable of activating peroxisome proliferator-activated receptor γ co-activator 1 α (PGC-1 α) transcription via AMPK activation [68]. In turn, H_2_O_2_ induces glucose uptake, but this is completel independently of AMPK activation [69]—apparently by activating a phosphatidylinositol 3-kinase (PI3K)-dependent mechanism [67,68].

NOX4 has been related to the physiological adaptation to acute exercise. Particularly, NOX4 deletion results in impaired glucose and fatty acid oxidization and decreased mitochondrial uncoupling protein 3 (UCP3) protein expression in response to acute exercise [70]. During muscle fiber electrical stimulation or insulin stimulation, NOX2 activation also increases H_2_O_2_ production, and the inhibition of this isoform decreases glucose uptake in myotubes [71]. Moreover, a specific source of ROS has been linked to exercise-stimulated glucose uptake, where H_2_O_2_ from NOX2 could be primarily responsible for GLUT-4 translocation during moderate-intensity exercise [72]. Moreover, H_2_O_2_ enhances GLUT4 translocation in skeletal muscle cells, independently of insulin, and is reduced by antioxidants and NOX2 inhibition. RyR-mediated Ca^2+^ release and IP_3_R-mediated mitochondrial Ca^2+^ uptake, alongside the canonical pathway, jointly promote glucose uptake in response to insulin [73]. Indeed, piperine administration induces intracellular Ca^2+^ and ROS generation via TRPV1, leading to calcium/calmodulin-dependent protein kinase kinase β-dependent AMPK phosphorylation and, consecutively, GLUT4 translocation and AMPK activation in L6 myotubes [74]. On the other hand, our research group has found an important connection between the electrical stimulation of skeletal muscle and the release of ATP into the extracellular medium. This ATP release appears to be also mediated by NOX2 [62]. Also, extracellular ATP is capable of increasing glucose uptake in skeletal muscle in response to exercise by PI3K activation [70]. All these findings point to a role of extracellular ATP in activating NOX2, which could modulate glucose uptake. Therefore, NOX2-dependent ROS production is a crucial mechanism for increasing muscle glucose uptake during exercise.

### 3.3. Mitochondrial ROS Associated to Physical Activity

Another hormetic response to physical activity is the induction of mitochondrial biogenesis, also called mitohormesis, which is mainly induced to enhance respiratory capacity and endurance [60]. Thus, exercise-induced mitochondrial ROS production might have an adaptive role at different levels, including the spread of energy between different pools of mitochondria, the fusion/fission state, and the status and location of mitochondrial Ca^2+^.

ROS production depends on exercise characteristics regarding the previous training level, intensity, and duration. Exercise can be moderate or intense, and it could be aerobic/endurance or resistance. Mitochondria and NADPH oxidases are the main sources of ROS production in aerobic exercise. Considering that complexes I and III of the electron transport chain are the main sites of mitochondrial superoxide production [75], the logical thinking is that increased ROS generation by contractile activity is associated with increased mitochondrial activity, increasing the superoxide generation dramatically by skeletal muscle during aerobic contractions [76]. However, during aerobic exercise, there is a reduction in mitochondria-derived superoxide anion formation compared to at rest, which is attributed to changes in the redox status of muscles shifting towards a more oxidative status to contraction. Indeed, when isolated mitochondria were incubated in a medium mimicking the cytosol of rat skeletal muscle during mild or intense aerobic exercise, the total rate of H_2_O_2_ production decreased to about 25% or 15%, respectively, of the rate in a medium mimicking rest [77]. On the other hand, a cytosolic source of ROS in aerobic exercise is NADPH oxidase, and interplay between the NADPH oxidases and mitochondrial ROS sources has been proposed, probably as an adaptive mechanism involved in slow and fast twitch-plasticity [2].

In aerobic exercise, the mitochondrial ratio of reduced NADH to oxidized NAD+ (NADH/NAD+) decreases. Low NADH/NAD+ ratio is associated with a decrease in the release of free radical oxygen species via complex I of the electron transport chain [78]. However, high-intensity aerobic exercise adversely affects mitochondrial function together with a decline in glucose tolerance [79]. Moreover, endurance exercise impacts the mitochondrial life cycle because Drp1 deficiency impairs muscle endurance and running performance and alters exercise-induced muscle adaptations [80]. On the other hand, resistance training enhances muscle strength and hypertrophy but may decrease mitochondrial volume due to hypertrophy outpacing mitochondrial biogenesis. Despite this reduction, the evidence does not support a net loss of mitochondria, and specific aspects of mitochondrial function may be improved with resistance training [81]. In aging rats, quadriceps femoris muscle mitochondria exhibit Ca^2+^ accumulation, ROS increase, reduced mitochondrial membrane potential, and decreased fusion protein levels. However, resistance training effectively mitigates these alterations [82]. Alternatively, in humans, resistance training for moderate periods, such as 12 weeks, does not significantly affect mitochondrial ROS production [83]. But resistance training induces mtDNA shifting; mitochondrial fusion via Mfn1, Mfn2, and Opa1; and mitochondrial biogenesis markers, contributing to improved mitochondrial function [32,84,85].

The transient increase in ROS production resulting from fission events during exercise may act as a signaling mechanism, triggering adaptive responses in the body, and orchestrating the beneficial adaptations that arise from physical activity.

## 4. Conclusions

Many questions remain when studying both oxidative metabolism and ROS production in skeletal muscle. The aim of this review is to bring attention, on one side, to the beneficial role of ROS in skeletal muscle physiology as opposed to their deleterious effects. On the other side, the particular distribution of skeletal muscle mitochondria and their different composition regarding key metabolic enzymes, together with the evidence pointing to both fusion events and fast sequential changes in mitochondria membrane potential, allow us to propose an energy-saving model consisting of distinct compartments where mitochondria membrane potential is generated in one compartment and, when needed, is transferred to a second compartment where ATP synthesis can take place. This distribution of functions saves energy and minimizes ROS production in muscle cells. It is important to note that mitochondria dynamics, i.e., the balance between mitochondria fusion and fission, is essential to this process and that normal muscle functioning depends on the capacity of mitochondria to fuse and fission at the right moment. This may explain why alterations in the mitochondria dynamics machinery have been reported in conditions (obesity and aging sarcopenia) in which muscle function is altered.

## Figures and Tables

**Figure 1 antioxidants-12-01624-f001:**
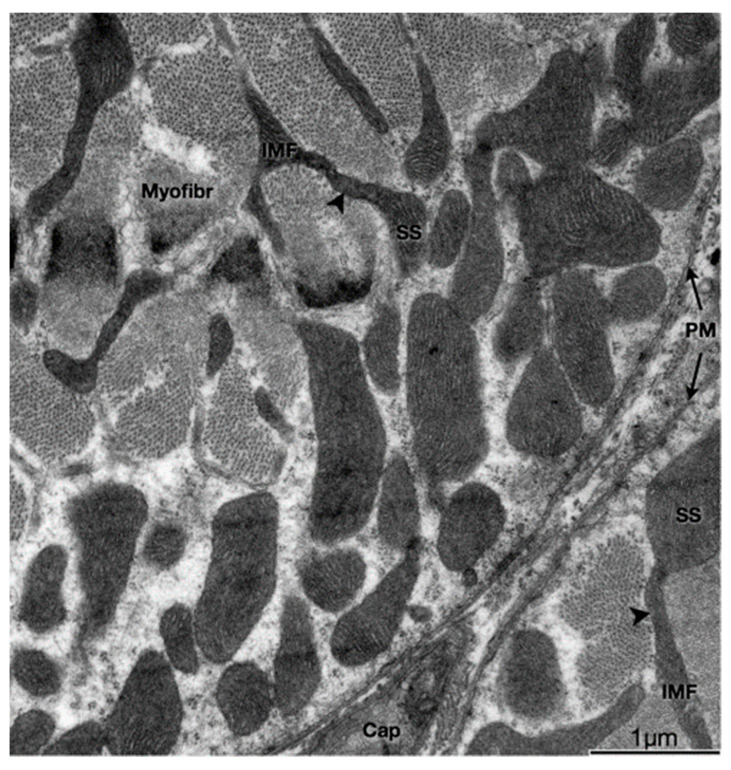
Physical interactions between SS and IMF mitochondria. Transmission electron micrograph of myofibers in the transverse plane. SS and IMF mitochondria are distinct organelles. Some SS and IMF mitochondria form continuous organelles (arrowheads) that coexist in both subcellular compartments. SS, Subsarcolemmal; IMF, intermyofibrillar; PM, plasma membrane (sarcolemma); Myofibr, myofibrils; Cap, capillary (Reproduced with permission from M. Picard (J. Appl. Physiol., published by American Physiological Society, 2013) [7]).

**Figure 2 antioxidants-12-01624-f002:**
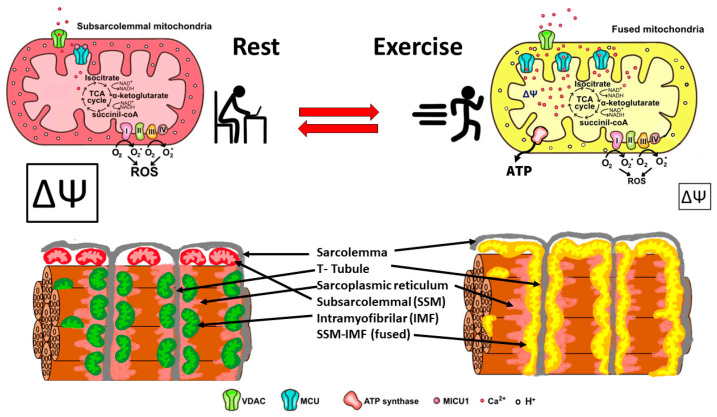
Energy efficient distribution of mitochondria that saves energy and minimizes mitochondria ROS production. Under resting conditions, skeletal muscle has different pools of mitochondria, SSM have a high mitochondrial potential (ΔΨ) and a basal rate of ROS generation from superoxide anions generated by the electron transport chain. Thus, the respiratory chain may allow the “leakage of electrons” and generate ROS. IMF mitochondria have a high expression of MICU1, which prevents calcium from massively entering the mitochondrial space. A condition of exercise implies that the SSM and the IMF mitochondria fuse, generating a network characterized by a lower density of MICU1, a high density of MCU, and an extended proton gradient (enough ΔΨ) needed to generate ATP and to minimize ROS generation. Colors of mitochondria represent diversity (red and green), and yellow represents fused mitochondria.

**Figure 3 antioxidants-12-01624-f003:**
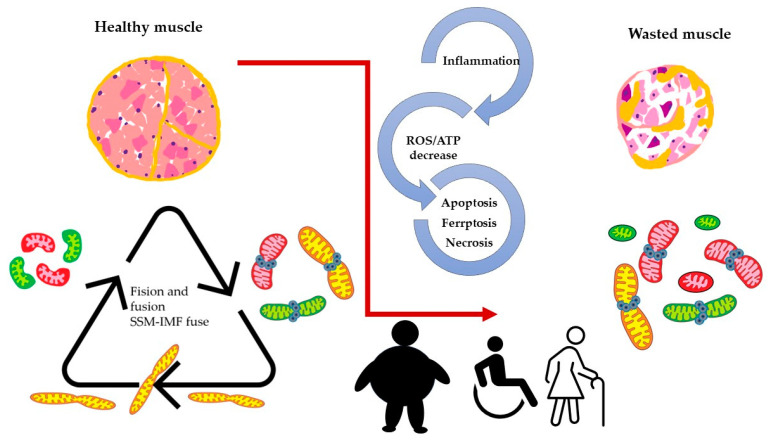
Altered dynamics and fusion capacity of mitochondria in disease. A common factor in many pathological conditions affecting skeletal muscle is an imbalance in mitochondrial dynamics. This imbalance is both a cause and consequence of inflammation, increased ROS production, and cell death, leading to muscle deterioration characterized by dysfunctional mitochondria. Different colors of mitochondria refer to SSM, IFM and fused mitochondria as pictures in Figure 2.

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
