# Peer review of "Energy (and Reactive Oxygen Species Generation) Saving Distribution of Mitochondria for the Activation of ATP Production in Skeletal Muscle"

_antioxidants, 2023, doi:10.3390/antiox12081624_

Round 1
Reviewer 1 Report
Dear authors,
your manuscript could be a masterpiece in the field of muscle bioenergetics. I appreciated the description of how the interactions of mitochondria located in different parts of muscle fiber can modulate all the contraction machinery by saving energy. The role of both SSM and PLM is well described, as well as the mechanisms of mitochondrial ROS production and modulation.
However, a bit of revision in the English Language could increase the understanding of your manuscript, as reported below.
English is quite good and understandable.
However, the Title should be rearranged and simplified to be more appealing and immediately readable.
Rearrange the sentence in line 80 as bad word repetitions are present (increase...increases)
line 207 - dynamis???
the sentence in lines 81-85 is not so clear to me, as well as the sentence in lines 87-89.
Pay attention to word repetitions throughout the entire manuscript.
Author Response
Reviewer 1
Dear authors,
your manuscript could be a masterpiece in the field of muscle bioenergetics. I appreciated the description of how the interactions of mitochondria located in different parts of muscle fiber can modulate all the contraction machinery by saving energy. The role of both SSM and PLM is well described, as well as the mechanisms of mitochondrial ROS production and modulation.
However, a bit of revision in the English Language could increase the understanding of your manuscript, as reported below.
Comments on the Quality of English Language
English is quite good and understandable.
However, the Title should be rearranged and simplified to be more appealing and immediately readable.
Rearrange the sentence in line 80 as bad word repetitions are present (increase...increases)
line 207 - dynamis???
the sentence in lines 81-85 is not so clear to me, as well as the sentence in lines 87-89.
Pay attention to word repetitions throughout the entire manuscript.
We thank the reviewer for the positive criticism and encouragement; we changed the title and made all the suggested English language changes.
Reviewer 2 Report
Review
The topic of this review is important and countless research groups are working to find out the role of ROS in skeletal muscle. The review proposes a new model how ROS production by mitochondria depends on exercise. The paper is well written however I have some major and minor comments what should be addressed.
Major points:
The authors start the review with the classification of different type of mitochondria. To visualize them a figure showing electro micrograph of a skeletal muscle section would be helpful for the readers.
I have some suggestion to improve Figure 1. First of all, I would insert a bidirectional arrow between the cartoon of REST and EXERCISE. This can propose that the process can go on backward mode too. I also suggest to modify the image of the fiber. In the present form IMF mitochondrias look as surface locating ones. The Authors can modify and increase a bit the tube-like fiber to similar what is in the following classical paper by Fawcett McNutt, J Cell Biol, 1969, 42, 1-45, Figure 30. Please explain in the figure legend what do you want to say with the bigger and smaller ΔΨ. After modifying the figure, Section 2.2 should be extended with explanation of the backward action. It also should be explained in this section why the Authors think that the fusion of IMF and SSM is the only way to change the ΔΨ of IMF? IMFs are in the close proximity of the T-tubules which can also transmit the membrane potential changes to them.
I have some concern also about Figure 2. Actually it is not cited in the text. What represents the color code of the mitohondrias? What is the meaning of the cartoon of foots with an arrow?
The Authors used a lot of abbreviations and about 50% of them is not defined. I would suggest to prepare a LIST of ABBREVIATIONS to overcome this discrepancy.
In section 3.1 the Authors discuss the effects of different concentration of H2O2 in skeletal muscle without giving its physiological concentration. Its range for rest and during exercise would be helpful referenced here.
Also in section 3.1 the Authors discuss the effects of H2O2 on CaV1.1. I miss a similar description of oxidation of RyR and SERCA. Both of them were shown to be reactive to H2O2.
Finally, I found Section 3.3 very short compared to the previous sections. The Authors can extend this paragraph with references presenting different levels of ROS as a consequence of different level of exercise.
Minor points:
Please insert a space between the text and the reference square bracket ([) where it is missing.
Please clarify which membrane potential you mean in row 22.
Please define triad in row 56.
Please correct RyR2 to RyR1 in row 64.
Please correct the sentence in row 74 as follows: “… and has different membrane potential.”
Please correct ER to SR in row 100.
Please correct flexus to flexor and give a reference in row 118.
Please rewrite the sentence in row 138-140. It is not clear for me.
Please correct the text in row 180 as follows: “The model in Figure 1 shows”
Please define ETC in row 196.
Please decide how do you type Fis1 and Drp1 (see row 228 and 241).
Please correct roles to role in row 250.
Please type DMD in italic if this a gene in row 251.
Please clarify what do you mean “In the first mechanism” in row 281.
Author Response
Answer to Reviewer 2
The topic of this review is important and countless research groups are working to find out the role of ROS in skeletal muscle. The review proposes a new model how ROS production by mitochondria depends on exercise. The paper is well written however I have some major and minor comments what should be addressed.
We thank the reviewer for his careful work and many suggestions. We addressed all of the points raised.
Major points:
The authors start the review with the classification of different type of mitochondria. To visualize them a figure showing electro micrograph of a skeletal muscle section would be helpful for the readers.
We included a new figure (now figure 1) of an electro micrograph from a muscle section.
I have some suggestion to improve Figure 1. First of all, I would insert a bidirectional arrow between the cartoon of REST and EXERCISE. This can propose that the process can go on backward mode too. I also suggest to modify the image of the fiber. In the present form IMF mitochondrias look as surface locating ones. The Authors can modify and increase a bit the tube-like fiber to similar what is in the following classical paper by Fawcett McNutt, J Cell Biol, 1969, 42, 1-45, Figure 30. Please explain in the figure legend what do you want to say with the bigger and smaller ΔΨ. After modifying the figure, Section 2.2 should be extended with explanation of the backward action. It also should be explained in this section why the Authors think that the fusion of IMF and SSM is the only way to change the ΔΨ of IMF? IMFs are in the close proximity of the T-tubules which can also transmit the membrane potential changes to them.
The figure was modified as suggested and section 2.2 was extended as suggested. We also explained that fusion is the only way that mitochondria membrane potential can propagate since there is no electrical continuity with other structures.
I have some concern also about Figure 2. Actually it is not cited in the text. What represents the color code of the mitohondrias? What is the meaning of the cartoon of foots with an arrow?
The cartoon was modified to indicate obesity and the color code of mitochondria(meaning diversity) was explained.
The Authors used a lot of abbreviations and about 50% of them is not defined. I would suggest to prepare a LIST of ABBREVIATIONS to overcome this discrepancy.
A list of abbreviations was included
In section 3.1 the Authors discuss the effects of different concentration of H2O2 in skeletal muscle without giving its physiological concentration. Its range for rest and during exercise would be helpful referenced here.
A reference for H2O2 concentrations was added.
Also in section 3.1 the Authors discuss the effects of H2O2 on CaV1.1. I miss a similar description of oxidation of RyR and SERCA. Both of them were shown to be reactive to H2O2.
Finally, I found Section 3.3 very short compared to the previous sections. The Authors can extend this paragraph with references presenting different levels of ROS as a consequence of different level of exercise.
Section 3.1 was significantly extended to include the points indicated.
Minor points:
All minor points indicated by the referee were corrected, either to clarify some sentences, to correct errors and to better define terms.
Round 2
Reviewer 2 Report
Review
The paper is improved sufficiently. I have just one major and some minor comments.
Major point:
The usage of IMF abbreviation is not consistent. Please check it in row 168, 205, 220, 221.
Minor points:
Thank you for the new Figure 1. Please cite it in the text. Please try to increase the visibility of the signs (SS, IMF, PMF, Cap) with different color of letters (i.e. white or yellow). Black letters are almost invisible.
Please decide that you use EC or E-C (coupling) in the text.
Please correct Ca2+ in row 42.
Please delete a dot at the end of row 106.
Please correct SSM to SS in row 166.
Please correct H2O2 in row 335 and 445.
Please correct IP3R in row 380.
Please delete the space after [65] in row 394.
Please rephrase “muscular depolarization” in row 407.
Please define the meaning of “O2•-“ in row 452.
Please correct Resistance in row 456.
Author Response
We thank the reviewer for the thorough revision; all the points were addressed and we corrected all the mistakes the reviewer noticed. Figure 1 is now cited in the text and we improved size and contrast to make the text more visible. The use of IMF abbreviation is now consistent throughout the text.